# Two Distinct Molecular Types of Phytochrome A in Plants: Evidence of Existence and Implications for Functioning

**DOI:** 10.3390/ijms24098139

**Published:** 2023-05-02

**Authors:** Vitaly A. Sineshchekov

**Affiliations:** Biology Department, MV Lomonosov Moscow State University, 119234 Moscow, Russia; vsineshchekov@gmail.com

**Keywords:** plants, phytochrome A, polymophism, fluorescence, photoreaction, photoresponse

## Abstract

Phytochrome (phy) system in plants comprising a small number of phytochromes with phyA and phyB as major ones is responsible for acquiring light information in the red—far-red region of the solar spectrum. It provides optimal strategy for plant development under changing light conditions throughout all its life cycle beginning from seed germination and seedling establishment to fruiting and plant senescence. The phyA was shown to participate in the regulation of this cycle which is especially evident at its early stages. It mediates three modes of reactions—the very low and low fluence responses (VLFR and LFR) and the high irradiance responses (HIR). The phyA is the sole light receptor in the far-red spectral region responsible for plant’s survival under a dense plant canopy where light is enriched with the far-red component. Its appearance is believed to be one of the main factors of plants′ successful evolution. So far, it is widely accepted that one molecular phyA species is responsible for its complex functional manifestations. In this review, the evidence of the existence of two distinct phyA types—major, light-labile and soluble phyA′ and minor, relatively light-stable and amphiphilic phyA″—is presented as what may account for the diverse modes of phyA action.

## 1. Introduction

In photobiology, investigations of the phytochrome system of plants are one of the most interesting and highly important problems both in fundamental and practical terms. The key role of the photoreceptor phytochrome is observed both in the individual development of the plant and in the evolution of higher plants in general. It regulates seed germination and transition from scoto- to photomorphogenesis, induces flowering and fruiting and triggers the processes of senescence [1]. The evolution of higher plants is largely associated with the appearance of a photoreceptor capable of perceiving light information under conditions of the dominance of the far-red and infrared areas of the solar spectrum characteristic of spaces under dense forest canopy [2,3].

Investigations of phytochrome are obviously important for artificial light culture [4,5]. It is also becoming evident that they may be useful for nanotechnological purposes. Phytochromes are considered to be effective optogenetic tools and fluorescent reporters working in the near infrared spectral region transparent for biological tissues [6,7]. It can serve as a molecular trigger, which is turned on and off by light of different spectral compositions, and as a nanoparticle and drug delivery carrier [8].

The discovery of the photoreceptor in the middle of the last century is associated with the detection of red (R)-induced/far-red (FR)-reversible physiological effects in plants (such as induction of germination, flowering, and fruiting) and R/FR reversible changes in the absorption spectra of plant tissues [9,10]. Research in these two main areas—photophysiology of R/FR reversible processes and physicochemical study of the photoreceptor in vitro—led, first, to the characterization of the photoreceptor as a molecule—and, second, to the phenomenological description of particular photoregulation plant reactions and categorizing them by their dose requirement into distinct photoresponse modes. Later on, with the advancement in molecular biology and genetics, the mechanism of the light signal transduction from phytochrome has been described in its main details—localization of the photoreceptor in the cell and intracellular trafficking was elucidated, and its signal transduction partners were determined [11].

Very schematically, the chain of events leading from the acquisition of the light signal through its transduction to realization can be described as follows. Phytochrome, a dimeric protein with a linear chromophore, bilin, upon absorption of light quanta undergoes photoisomerization reaction converting the initial red-light absorbing form Pr into the first stable at low temperatures (T) photorpoduct lumi-R, and following its dark transformations into the far-red absorbing physiologically active form Pfr. Pfr can return into Pr via a photochemical branch or thermally, thus completing the Pr↔Pfr photocycle [12,13,14]. This activation of phytochrome with the appearance of Pfr takes place in the cytoplasm, where the photoreceptor is synthesized in the Pr form in the dark, and leads to its transfer into the nucleus [15,16,17,18,19]. In the nucleus, Pfr deactivates phytochrome interacting factors (PIFs) blocking photomorphogenesis and activating factors (HY5), initiating it. Activation of photoresponsive genes leads to numerous phenomenological growth and development reactions [20,21,22,23,24]. Although most of the examined phyA actions require the nuclear import of the photoreceptor, cytoplasmic fraction of phytochrome is likely to be engaged in several biophysical and biochemical events [25,26,27,28,29]. Depending on the spectral composition of actinic light, their dose and pulsed or constant mode of illumination, photophsyiological reactions are divided into the very low fluence responses, VLFR, the low fluence responses, LFR, and the high irradiance responses, HIR [30,31,32,33,34,35,36].

For quite a while, it was widely accepted that plant’s responses initiated by light in the red and far-red spectral light were mediated by one molecular phytochrome species. Subsequently, however, there were a number of observations pointing to the heterogeneity of phytochrome in the cell and suggesting that there might be distinct phytochromes that can monitor discrete elements of the light environment [37,38]. These include the detection of phytochrome pools that differ in (a) light lability—light-labile type 1 and light-stable type 2 phytochromes [39], (b) immunochemical properties [40,41], and (c) fluorescent and photochemical parameters [42,43].

Compelling evidence for the existence of distinct phytochromes was obtained by the discovery of a divergent family of phytochrome genes [1,37,38,44,45,46,47,48]. This implied that different phytochrome types might have unique photoregulatory roles. The latter was demonstrated with the use of photomorphogenic mutants and phytochrome-overexpressing transgenic plants—the VLFR and HIR were attributed to phyA, while the LFR, was to phyB. The phyA may also mediate the LFR. The light signal transduction from phyA and phyB proceeds, with different dynamics of the process, via separate chains and interacting partners [35,49,50,51,52].

Data are being accumulated that the complex functions of phyA in plants are connected, at least partially, with the molecular diversity of the photoreceptor. In one plant species, several phyAs—different *PHYA* gene products—may be present with distinct functions, an effect termed subfunctionalization/neofunctionalization [3,53,54,55,56,57]. Upon phyA gene duplication during plants′ evolution, they may undergo structural modifications that considerably change their phenomenological properties and functions. Lin et al. [58] have shown, in particular, that the regulation of soybean flowering is mediated by the products of the two *PHYA* genes, phyA3 and phyA2, with distinct light stability and distinct roles in this process. The phenomenon of the *PHYA* gene duplication with structural modifications of their product phyA cannot account, however, for the specificity and complexity of the phyA functioning in the majority of monocots and dicots. In this review, I am discussing a more general case—the existence of post-translational modification(s) products of one and the same phyA gene in a plant possessing distinct photophysiological functions. This approach predetermined the choice of the material for the analysis and citation. For general comprehensive information on phyA properties and mechanisms of action, the reader is addressed to a number of excellent reviews [23,52,59,60,61,62,63].

## 2. Phytochrome A Heterogeneity in the Cell: Chemically Distinct Species and Conformers

The means of phytochrome investigation in situ is limited because of very low concentration in plant tissues (<10^−7^ mol/L) and the presence in them of photosynthetic pigments interfering with measurements. In fact, there was only one spectroscopic method—difference absorption spectroscopy—that allowed direct measurements of pigment’s R/FR—induced difference spectrum, and, based on that, reconstruction of its absorption spectra in the Pr and Pfr forms [64,65]. In our group, the fluorescence of phytochrome *in-planta* was detected and a sensitive and informative method of phytochrome investigation in its natural state in plant cells was developed [42,43,66]. Fluorescence emission and excitation spectra of Pr (Figure 1) and also of the photoproduct lumi-R were recorded in etiolated plant tissues of monocots and dicots in the dark-adapted state when all the pigment is in the initial Pr form, and in the state of Pr photoequilibrium with lumi-R at cryogenic T (Tc) (77–85 K) and with Pfr at ambient T (Ta) (for procedure of the measurements see Figure 2). Pfr was shown to be lacking fluorescence even at liquid helium T (4 K) [14]. These spectra, temperature dependencies of the Pr fluorescence yield, and fluence time-response curves of its changes in the Pr→lumi-R photoreaction allowed for the determination of several phytochrome parameters—position of the fluorescence emission and excitation (absorption) maxima, activation and kinetic parameters of the Pr photoreaction and fluorescence quenching, relative phytochrome content in the sample, proportional to the Pr fluorescence intensity, and the extent of the Pr conversion into lumi-R at (Tc) and into Pfr (at Ta). All these parameters were shown to vary depending on plant species and genotype, its developmental state, organ/tissue used, growth conditions, and environmental factors, light pretreatment, in particular (Figure 3). This phenomenon was interpreted as a manifestation of the heterogeneity of phytochrome in vivo—of the existence of at least two distinct phytochrome species, Pr′ and Pr″, characterized by different spectroscopic and photochemical parameters (Table 1). Most profound changes were observed in the ability of Pr to undergo photoconversion into lumi-R, measured as a relative Pr fluorescence decay after saturating red (R) preillumination at Tc (parameter γ_1_ that varied from approx. 0 to 0.5). This allowed for the determination of the Pr′/Pr″ proportion in the sample from the experimental γ_1_ value as described in [67]. The [Pr′] and [Pr″] showed great variations depending on the above conditions, in particular, with total phytochrome content, P_tot_, during the seedling’s development and upon its actinic red preillumination (Figure 4). From these dependencies, Pr′ can be characterized as a major and light-labile species, whereas Pr″ as a minor, saturable, and relatively light-stable (Table 1).

With the discovery of the two major phytochromes A and B—the major and light-labile phyA (type 1) and minor and light-stable phyB (type 2) [43]—, it was tempting to assign the Pr′ and Pr″ species to phyA and phyB, respectively. The picture proved, however, to be more complex than that. It was found that mutants lacking phyB—monocots (rice) and dicots (cucumber, *Arabidopsis*, pea)—contained practically the same amounts of Pr′ and Pr″ as the respective wild types suggesting that both Pr species belong to phyA differing presumably by post-translation modification (designated as phyA′ and phyA″) (Figure 5). Later on it was shown that phyB reveals properties close to those of phyA″ and belongs to the same Pr″ type [72]. The heterogeneity was also observed in the case of phytochrome of lower plants (*Adiantum* phy1) [73]. To make the picture even more complex, it was found that phyA′ (photochemically active at Tc) is itself heterogenous comprising distinct conformers. This was shown by the complexity of the fluence time-response curves of the Pr→lumi-R conversion and its temperature dependence [74]. Thus, it was concluded that the system of phytochromes in plants possesses three levels of complexity—(1) different gene products, (2) post-translationally modified products of one and the same gene, and (3) conformers within a distinct phytochrome species (see reviews [14,75,76]).

The notion of the heterogeneity of phyA is supported by the investigations of the cyanobacterial phytochrome Cph1 which is considered a suitable model of plant phytochromes [60]. In our works, absorbance and fluorescence characteristics of Cph1 were shown to be similar to those of plant phytochromes. The fluorescence intensity of Cph1 showed steep temperature dependence suggesting that the fluorescence decay is a thermally activated process. Fluorescence measurements also revealed phototransformation of Pr only at T above 150–160 K. This attributes them to the Pr′′ type inactive at Tc (similarly to phyA′′ and phyB and in contrast to phyA′ which converts into lumi-R at Tc with the extent up to 0.5). Two species with distinct fluorescence and photochemical characteristics and *E*_a_ but with relatively equal yields of the photoconversion at ambient T were also detected in the case of cyanobacterial Cph1 which are considered to be distinct conformers of the pigment [81]. Time-resolved and structural investigations of Cph1 in recent years have confirmed the notion of the heterogeneity of the ground state Pr population as a source of the complex kinetic and energetic processes in the phytochrome molecule. Thus, this phenomenon is not likely to be a unique feature of phyA (see discussion in [82]).

## 3. Photochemical and Structural Characterization of the Two phyA Types

The fluorescent and photochemical diversity of the phytochrome species can be interpreted in the framework of the scheme of the initial photoprocesses in photoisimerizing pigments (Figure 6) [70,81] (for detailed discussion, see reviews [14,82]). According to it, the photoisomerization, a single rotation around C15=C16 double bond in the excited Pr* state, has two major features which allow for the interpretation of the two phenomenological facts—first, the ability and inability of the two phyA species, respectively, to undergo the photochemical Pr→lumi-R conversion at Tc, and second, their almost identical quantum yields and extent of the Pr→Pfr phototransformation at Ta. These are, respectively, the existence of the activation barrier *E*_a_ in the excited Pr state (Pr*) for the photoreaction and of the orthogonal prelumi-R “hot” ground state (real or virtual), where the branching of the photochemical routes takes place—direct productive prelumi-R→lumi-R or reverse unproductive prelumi-R→Pr. In this branching, the reverse process dominates thus lowering the yield of the Pr→Pfr photoconversion which is evaluated to be around 0.13–0.15 [12,13,14,60]. In line with this energetics point of view, phyA″ (inactive at Tc) is characterized by a high *E*_a_ of about 20–30 kJ/mol, whereas for phyA′ (photoactive at Tc) it is much lower—hundreds J/mol. The conformers within phyA′ are also distinguished by *E*_a_ [74]. Similarly, Cph1 comprises two subspecies—the minor, moderately fluorescent with *E*_a_ = 12.5–17.5 kJ/mol and the major, weakly fluorescing with *E*_a_ = 3.0–6.5 kJ/mol. Both of them are inactive at Tc but reveal practically equal photoactivity at Ta [81].

The energy barrier *E*_a_ and the pattern of the branching of the conversion pathways at the prelumi-R point need to be interpreted in terms of the phytochrome molecular structure and chromophore-apoprotein interactions. Comparative analysis of the fluorescent and photochemical data of wild-type phyA and Cph1 and their deletion mutants allowed localization of the site in the molecule responsible for the initial photochemical events. It was found that the N-terminal photosensory module (PSM) is autonomous in this regard. Complete deletion of the C-terminal module does not affect the properties of phytochrome A suggesting that the Pr′/Pr″ (that is, phyA′/phyA″) ratio is not changed [83]. Similarly, there is practically no difference between Cph1 and its photocensor domain Cph1Δ2 [81,84]. On the contrary, the deletion of the 10 kDa N-terminal segment in phyA (NTE) (Δ7–64 phyA) was critical for the formation of Pr′ (phyA′), so that all the pigment was represented by the Pr″ (phyA″) pool [83]. Thus, PSM and NTE (the latter, in the case of phyA) are the major players in the photoprocesses, and dimerization has no effect on them, given that the C-terminaly deleted pigments are monomers.

The fact that the N-terminally truncated phyA is lacking phyA′ and is present only in the phyA″ state may raise the question if the minor phyA″ species is a product of degradation of the major phyA′, i.e., partially degraded 118-kDa and/or 114-kDa phytochrome. This is, however, not the case because full-length *Arabidopsis* and rice (*Oryza*) phyA expressed in yeast *Pichia pastoris* [85] and *Arabidopsis* phyA expressed in *P. pastoris* and *E. coli* [71] (assembled in vivo with phycocyanobilin (PCB) or phytochromobilin (PΦB)) all belong to the Pr″ phenomenological type and are similar to or identical with the phyA″ in plants. This was shown by (1) the fluorescence emission spectra, (2) the temperature dependence of Pr fluorescence intensity and activation energy of fluorescence decay, and (3) the extent of photoconversion of Pr into photoproduct lumi-R (γ_1_) or far-red–light–absorbing form (Pfr) (γ_2_) (Figure 7). The data thus demonstrate that the low-abundance–fraction plant phyA (phyA″) comes from the same gene as the major (phyA′) fraction. They were also interpreted as an indication that the formation of phyA′ is a property of the higher plants and that *P. pastoris* and *E. coli* do not possess the mechanism to convert phyA″ into phyA′.

The site responsible for the phyA differentiation can be localized in the NTE even further—the serine-to-alanine substitution of the first 10 serines (of rice phyA expressed in phyA-less *Arabidopsis*) produced the same effect—the lack of the phyA′ pool with all the pigment present as phyA″ (Figure 8) [86]. This is yet another argument for the fact that phyA″ is a full-length pigment. Since phyA is a phosphoprotein the effect of this substitution suggested that the phyA modification could be serine phosphorylation. More specifically, (oat) phyA autophosphorylates at Ser8 and Ser18 (in the Pr and Pfr states) (see review [64]), and it was tempting to assume that this process may account for the phyA differentiation. The notion of phosphorylation was also supported by the observation that the treatment of the pigment in vitro by calf phosphatase shifts the proportion of the two pools towards phyA″ [14]. Moreover, the involvement of phosphorylation in the phyA differentiation is pointed out by the fact that the lack of phytochrome kinase substrate 1 and 2 (PKS1 and PKS2) causes a shift in the phyA pools′ ratio toward phyA″ (Figure 9) [87]. At the same time, there are data that contradict this concept. First, Ser8 and Ser18 are dispensable for the phyA″-into-phyA′ conversion. This is evidenced by the lack of the effect of the Ser8Ala and Ser18Ala substitutions on the phyA′/phyA″ balance [86]. Besides, the treatment of etiolated maize seedlings by okadaic (OA) and cantaridic (CA) acids (in stems) and by NaF (in roots) shifting the phosphatase/kinase equilibrium toward the former results in the increase in the phyA″ proportion, although one could expect to get more phyA′ at the expense of phyA″ (see [71,88] and discussion of this issue below). However, the option that some other serine(s) in the NTE serine cluster besides Ser8 and Ser18 is phosphorylated to form phyA′ from phyA″ is still open, and the nature of the mechanism of the phyA′/phyA″ interconversion needs further investigation.

Irrespective of the concrete mechanism of the phyA diversification, it is clear that modifications of the NTE terminus greatly affect the photophysical and photochemical properties of the two phyA pools most likely through alterations in the conformation of the chromophore and its environment and changes in the strength and character of its interaction with the apoprotein. The activation barrier *E*_a_ critical for the Pr→lumi-R photoreaction should relate to the energy of the D-ring bonds fixing it in the chromophore pocket. During D-ring rotation these H bonds are to be broken (Figure 10) [81]. The energies associated with hydrogen bonds are in the range 6–30 kJ/mol [89] which fits well into the magnitudes of *E*_a_ in plant phytochromes (from hundreds J/mol to 30 kJ/mol depending on the phyA pools [14]) and Cph1 (3.0–6.5 and 12.0–17.5 kJ/mol for the two Cph1 species [81]) (see above). Thus, the attribution of the *E*_a_ barrier to the H bonds of the D-ring fixing it in the chromophore pocket sounds quite reasonable.

Investigatons of Cph1, as an adequate model of plant phytochrome [60,82], allows for an explicit structure–function analysis of its molecule. Structural experiments reveal several critical points of interaction of the chromophore with its protein surrounding in Cph1. The C19 carbonyl oxygen of the D-ring forms a hydrogen bond with the nearby histidine His290 and a hydrogen bond between its pyrrole nitrogen N24 and a water molecule (Figure 10). The two highly conserved histidine residues, His323 and His372, are also involved in determining the conformation structure of the chromophore [91]. Of particular importance is the conserved tyrosine Tyr263 residue directly interacting with the D-ring [81,92]. Experiments with Cph1Δ2 and its mutants (Tyr263Phe, Tyr263His and Tyr263Ser) have shown that the chromophore in the WT is less twisted than in the mutants. *E*_a_ for the photoreaction Pr→lumi-R is not affected by the mutations (3.0 vs. 2.5 kJ/mol in the WT and the mutants, respectively). However, the probability of the Pr→lumi-R path at the orthogonal prelumi-R state in the photoreaction is higher in the WT (see the scheme in Figure 6), and thus the quantum yield of the photoconversion is also higher. In contrast to the mutant Cph1Δ2 species, which are homogeneous, the WT Cph1Δ2 was found to be represented by two species with distinct fluorescence spectra and quantum yields but with similar quantum yields of the Pr→Pfr photoconversion [81]. This is due to the structural plasticity of the chromophore provided by Tyr263. The two Cph1Δ2 species probably differ by the degree of their protonation because the pKa value of the phenol group of the Tyr263 is low, and this directly influences the protonation of the chromophore [93]. This property of Tyr263 may explain why the appearance of the distinct Cph1 population correlates with the state of chromophore protonation [94].

The second important tyrosine residue near the D-ring, which contributes to the structural inhomogeneity of Cph1, is Tyr176 as revealed by ultrafast kinetic measurements of the photoprocesses in Cph1Δ [95] in agreement with the static measurements of the plant phytochrome [74] and Cph1 [81,92]. The structural heterogeneity of the Pr state in Cph1 is also associated with the input of the hydrogen-bonding networks and charge distribution patterns around the chromophore as shown on Cph1 [91] and on (oat) phyA [96], and of multiple side chain conformations for several residues near the chromophore [90,92].

Thus, judging by the data on Cph1, the structure of the chromophore pocket in plant phytochrome provides freedom of movement removing sterical hindrances for the chromophore photoisomerization (Figure 10). This chromophore plasticity is reflected in the appearance of distinct conformers of the pigment—prevailing structures with energetically favorable chromophore-apoprotein interactions. In phyA, besides the conformers (within phyA′), the character of the chromophore-apoprotein interaction and the photophysical and photochemical parameters are greatly affected by the chemical modification of phyA in the phyA″→phyA′ transition (possibly, via phosphorylation of the NTE), although the interaction of the NTE with the chromophore in the Pr form of phyA is much less pronounced than in the Pfr form [97]. The key role of the apoprotein matrix in determining the physicochemical phytochrome parameters is vividly observed when cyanobacterial phytochromes are compared with light-harvesting biliproteins [82]. In the latter, there is a rigid chromophore surrounding stiffly fixing the chromophore, reducing its freedom of torsional relaxation, and restricting its movements. This provides for the high fluorescence quantum yield of a pigment, a prerequisite for its being an efficient energy donor. The strategy of the molecular organization and chromophore-apoprotein interaction of the two groups of pigments—photoisomerizing and light-harvesting biliproteins—is discussed in the review [82].

## 4. The phyA Pools and the Problem of the Membrane-(Protein-) Association of Phytochrome

Investigations of membrane-bound phytochrome date back to the early years of its research. Although the fact that the pigment was clearly shown to be soluble in in-vitro experiments, there were several publications reporting its pelletabily upon extraction and associations with different subcellular structures [98,99,100]. Photoreversible redistribution of Pfr sequestering in the cytoplasm was demonstrated by Mackenzie et al. [101]. A bound phytochrome fraction was observed in oat cells [102]. In *Mougeotia*, a role for membrane-bound phytochrome in chloroplast movement was demonstrated by Haupt [103]. The possible existence of the physiologically active membrane-(protein-) bound fraction of phyA is pointed out by experiments on higher plants too (see below). Our experiments suggest that phyA″ possesses amphyphylic properties (an ambiquitous phyA type, i.e., having the ability to reversibly bind to subcellular structures [104])—it comprises both a water-soluble fraction and membrane-(protein-) associated fraction, in contrast to phyA′ which is only in the water-soluble state. In our experiments, both phyA′ and phyA′′ were found in the supernatant (from etiolated maize coleoptiles) [85] or *Arabidopsis* hypocotyls [105], whereas there was primarily or exclusively phyA′′ in the sediment (Figure 11). Furthermore, phyA′ disappears upon deep dehydration of etiolated tissues in the sediment [106], suggesting that phyA′ molecules need a water surrounding for their stability, and that in dry seeds phyA is likely to be present in the phyA′′ form. The properties of the presumably membrane-(protein-) associated phyA″ are close to those obtained in the sedimentation experiments in [107,108].

These changes in the hydrophobicity/hydrophilicity of phyA are likely to be the result of the overall structural rearrangement of the pigment upon its post-translational modification. We may hypothesize that the increased solubility of phyA′ is the result of the presumed phyA phosphorylation of serine(s) at the NTE terminus. The appearance in the molecule of a charged phosphoryl group(s) changes its overall conformation (as indicated by the drastic amendments of the energetics of the photoreaction) bringing about an increase in its polarity and, as a result, its hydrophilicity. Previously, it was shown that the N-domain contains the determinants for the differences in photosensory specificity and photolability between phyA and phyB [48]. According to [45], the differential nuclear import of the two phytochromes (see below) could result from the N-domain–dependent change of surface properties of the C-domain in terms of hydrophobicity and reactability. We may speculate that this could be as well applied to the hydrophilic phyA′ and amphiphilic phyA″. Thus, we have three possible variants of the phyA state in the cell—the water-soluble phyA′ and the two fractions of amphiphilic phyA″, water-soluble and aggregated (protein-associated), which may explain the diverse and multiple functional phyA manifestations.

## 5. The Two phyA Spices May Account for the Two Distinct Patterns of phyA Nuclear Speckle Formation

In the process of light signal transduction, both phyA and phyB are imported into the nucleus in a light-dependent manner [15,16,17,18,19]. For phyA, it is the VLFR and takes minutes whereas for phyB it is the LFR completed within 1–2 h [110]. The phyA in its Pfr form needs association with plant-specific proteins FHY1 (Far-red elongated Hypocotyl 1) and FHL (FHY1-like) to achieve nuclear import [25,26,111,112,113,114]. The phyA amino-terminal extension (NTE) domain mediates the formation of the agregates of phyA with its partners [115]. In the nucleus, phyA is localized to protein complexes known as photobodies (also speckles or spots); they are of two types—many small or a few large ones [17,18,110,116,117]. Photobodies serve as environmental sensors in plants, they are important for light, circadian, and temperature signaling [17,118,119,120,121]. According to Menon et al. [122], retention of phyA in the cytoplasm is important to suppress photomorphogenesis in the dark. This is pointed at by the fact that lines expressing constitutively nuclear-localized phyA (phyA-NLS-YFP—phyA fused to the nuclear localized signal NLS) are hypersensitive to red and far-red light and that its presence in the nucleus results in photomorphogenic development in the dark.

In the context of the existence of the two phyAs of interest is the fact that phyA lacking the 6–12 amino acids from its N-terminus (*Avena satva* (oat) Δ6–12 phyA-GFP expressed in *Arabidopsis* deficient in phyA) can form only one type of the complexes—many tiny spots [123]. This and the fact that the full-length oat phyA-GFP is represented by both types prompted us to investigate if this may relate to the existence of the two phyAs in the cell [124]. It was found that phyA-GFP possesses the same spectroscopic and photochemical properties as the native phyA and that it is represented in the cell by two phyAs—phyA′-GFP and phyA″-GFP. This suggests that the GFP tag does not affect the chromophore and its immediate protein surrounding and is in good agreement with the previous studies showing that phyA-GFP is a functional photoreceptor [17,110,115,123]. The fact that phyA-GFP is represented in the cell as phyA′-GFP and phyA″-GFP implies that both of them are potential participants of the light-induced nuclear–cytoplasmic partitioning, and that the two types of light-induced phyA nuclear speckle formation may be indeed connected with the existence of the two phyA species. This assumption is supported by the observation that Δ6–12 phyA-GFP, which forms only numerous tiny subnuclear speckles, is solely represented by phyA′-GFP [124]. The large speckles observed in the case of full-length phyA-GFP, besides the tiny speckles, may thus be associated with phyA″. This implies that the two phyAs may mediate different photoresponses or diverse response modes (see below). One may envisage alternative schemes for the intracellular localization of the phyA′ and phyA″ species: (i) both of them are present in each cell and (ii) there are two different groups of cells containing primarily one or the other phyA type. This needs direct experimental verification.

## 6. The phyAs Mediate Distinct Types of Photoresponses: The Major and Light-Labile phyA′—The VLFR, and the Minor and Relatively Light-Stable phyA″—The HIR

Particular photoresponses in plants initiated by phytochromes,—such as regulation of seed germination, hypocotyl growth, hypocotyl gravitropic orientation, cotyledon unfolding, hook opening, greening, light-harvesting chlorophyll a/b-binding protein gene (LHCP) expression, and flowering,—are categorized into three photoresponse modes based on the quantitative relationship between response and predicted levels of the far-red light absorbing form of phytochrome (for a review see [35,36,125]). These are the VLFR mode (observed under pulses of FR or very low fluences of pulsed or continuous R), the LFR mode (under the conditions of continuous or pulsed R), or the HIR mode (continuous FR). With the use of phyA and phyB *Arabidopsis* mutants, it was clearly shown that phytochrome A mediates both the VLFR and HIR and phyB, the LFR [49]. The VLFR and HIR modes were absent in the *phyA* mutant and normal in the *phyB* mutant, whereas, on the contrary, the LFR were present in the *phyA* mutant and lacking in the *phyB* mutant. Thus, the LFR differ from the VLFR and HIR at the level of the photoreceptor and there are different reaction partners of phyA and phyB. Moreover, a number of loci have been identified that affect differentially the VLFR and HIR, which were associated with signal transduction components downstream from phyA [125]. Additionally, the HIR and VLFR operate via different regions of target gene promoters [126]. Thus, a model is put forward in which phyA initiates two transduction pathways, the VLFR and HIR, involving different cells and/or different molecular steps, i.e., signaling downstream of phyA branches out into two cascades, depending on the mode of phyA light activation. The HIR action spectrum with the maximum beyond 700 nm is explained by the shuttle-like process of the phyA transport in the nucleus (see above) [127].

The detection of the two phyAs implies, however, that the two modes of phyA responses, the VLFR and HIR, may differ also at the level of the effector, i.e., the distinct phyA species can separately initiate the different responses. To verify this, we have carried out experiments on phyA-302 mutants (with the substitution Glu777Lys) deficient in the HIR but retaining normal VLFR [128]. This is interpreted by the authors as a result of the altered electrostatic environment of the peptide region, which could impair phyA interaction with signal transduction proteins. These mutants reveled normal nuclear translocation upon FR illumination but failed to produce nuclear speckles. It was speculated that the Glu-777 residue in the PAS2 domain is involved in the HIR signaling via interaction with a partner not necessary for the VLFR, and that this happens in the nuclear speckles. Fluorescence in-situ measurements have shown that the total phyA content reduced approx. two-fold but its fluorescence and photochemical properties remained the same as in the WT. This suggests that the proprotion of the two phyAs remains the same too. In the other words, the HIR deficiency in this case is not connected with the hypothetical lack of one or the other phyA pool. Also, the amino acid substitutions Arg194Val and Cys581Thr in pea phyA increased the deetiolation phenotype under FRc and Rc without violation of the phyAʹ/phyAʹʹ balance [129,130]. Impaired HIR was observed also in plants expressing phyA with the substitution R384K [131] although the state of phyAs in them remains unknown. Thus, mutations in the phyA molecule, which do not affect the formation of the phyA pools, have an impact on the manifestation of the different photoresponse modes, most pronounced in the case of the HIR. The fact that these substitutions, outside the NTE, do not affect the phyA′/phyA″ ratio is in line with the notion that the NTE is the site responsible for the phyA differentiation (see above).

Yanovsky et al. [49] genetically dissected the photoresponse modes using a polymorphism between ecotypes Landsberg *erecta* and Columbia. The VLFR (seed germination and potentiation of greening, hypocotyl growth inhibition and cotyledon unfolding in etiolated seedlings) was severely deficient in Columbia, whereas the LFR and HIR were normal. We have carrried out experiments to clarify if the observed polymorphism between Landsberg *erecta* and Columbia could be caused by violations in the state and relative content of the two phyAs [132]. Columbia ecotype had the same characteristics of phyA and the proportion of its phyA′ and phyA″ populations as those of Landsberg *erecta*. This implies that the absence in Columbia of the VLFR mediated by phyA is not connected with the state of the photoreceptor and that the defects in the light signal transduction are located downstream from it.

The above results on phyA mutants and on Columbia affecting the strength of the HIR and VLFR, which did not reveal violations in phyAs properties and proportion, do not exclude, nevertheless, the possibility that the two distinct chains of phyA signal transduction (i.e., the VLFR and HIR) could be initiated separately by phyA′ and phyA″. Indeed, quite a different situation was observed in the case of phyA mutants with substitutions or deletions at the NTE bringing about a steep decline or abscence of one or the other phyA pool. We have shown that mutant rice phyA (phyA SA with the first 10 serines substituted by alanines) overexpressed in transgenic *Arabidopsis* deficient in phyA or both phyA and phyB comprises primarily or exclusively the phyA″ pool (see [86] and above). According to [50], these transgenic plants with the mutated rice phyA were much more active in the HIR responses than in the VLFR and LFR—in promoting under constant FR (1) inhibition of hypocotyl elongation, (2) anthocyanin accumulation, (3) agravitropic growth, and (4) ‘FR-killing effect’ (lethality of seedlings grown under FR upon illumination with R or W light [133]). This indicates that phyA″ is primarily responsible for the HIR of these deetiolation processes. The phyAʹʹ was also found to be much more effective in germination induction than phyA′ [86]. In contrast, WT rice phyA, which was represented by both phyA′ and phyA″, was more active in (1) inhibition of hypocotyl elongation and cotyledon opening under pulses of FR light (VLFR), (2) ‘FR killing effect’ after FR-light pulses (VLFR), (3) inhibition of hypocotyl elongation and agravitropic responses under R (LFR). This VLFR activity may thus be connected with the presence of phyA′ since phyA SA comprising phyA″ revealed itself as a mediator of primarily the HIR (Figure 12). It is not clear if the LFR can be attributed to phyA as suggested by Kneissl et al. [50], since it was shown to be mediated by phyB [49]. If so, one may hypothesize that phyA″ is a more appropriate candidate for the LFR activity because it is closer by its properties to phyB (photochemically and by light-stability; see above) than phyA′.

A different picture is observed in the case of the truncated Δ6–12 oat phyA expressed in transgenic tobacco and *Arabidopsis*, which according to our data is represented primarily by phyA′ (see above and [124]). Casal et al. [123] have shown that this truncated phyA was as active as the full-length phyA for the VLFR of hypocotyl growth inhibition, cotyledon unfolding and blocking subsequent greening under white light in *Arabidopsis*. In transgenic tobacco, it was hyperactive in the VLFR of hypocotyl growth inhibition and cotyledon unfolding. In both the plant species, Δ6–12 oat phyA revealed a dominant suppression of the HIR in these regulation reactions. These data suggest that in these expression systems the VLFR are mediated by phyA′ (see Figure 12 and [134]). The fact that phyA′ and phyA″ form different types of speckles in the nucleus (see [124] and above) suggests that the distinct modes of photoreposponses they mediate (the VLFR and HIR, respectively) proceed via different signal transduction chains in agreement with [125,126].

Experiments with transgenic plants overexpressing their own or endogenous phyA have shown that the attribution of the photoresponse modes to the distinct phyA species may not be strictly fixed and is susceptible to the physiological context. We have followed the content of the two phyAs in two systems which were characterized by the modes of their photoresponses—transgenic wheat overexpressing oat phyA and acquiring the HIR (for growth inhibition, leaf unrolling, and anthocyanin formation), which was not characteristic of the WT wheat [135], and transgenic potato over- and underexpressing endogenous phyA, which showed accelerated and delayed HIR, respectively (for stem extension, leaf expansion, and hook opening of sprouts) [136]. In the transgenic wheat, phyA′ was the dominating species in the overexpressed oat phyA [137]. This suggests that it could be responsible for the altered phenotype of the transgenic wheat. Under these conditions, it is thus likely that the HIR is mediated by this overexpressed oat phyA′. In transgenic potato, the most dramatic variations were observed in the concentration of phyA′ (its content changed 40–60-fold in going from the under- to overexpressor, whereas [phyA″] changed only 4–6-fold [138]. It is tempting to associate the changes in the phenotype with the major phyA′ pool and to a lesser extent with the minor phyA″.

Thus, there is a seeming controversy between the attribution of the VLFR and HIR modes based on the overexpressors of the mutant phyA represented either by phyA′ or by phyA″ and overexpressors of the WT phyA comprising both the phyAs pools. To explain this, we may consider two opportunities. First, the conclusion based on the correlation between changes in the content and proportion of the two phyA pools in transgenic plants (primarily with phyA′) and modifications of their phenotype and photoresponse modes is erroneous. It may arise because of the presence of phyA″ in these lines. Even a relatively small increase in [phyA″] may account for the modified phenotype because a relatively moderate increase in [phyA] (two–three-fold) brings about the saturation of its action [139]. On the other hand, the attribution of the HIR to phyA′ can be correct for the system under investigation—transgenic wheat and potato—because phyA responses can be substantially modified in transgenic plants [123]. In [134], we speculated that phyA′ in overexpressors can acquire the properties and functions of phyA′′. This assumption is supported by the observation [137,140] that overexpressed phyA is relatively light-stable, which is characteristic of phyA′′. We may thus assume that under conditions close to physiologically normal ones, phyA′ and phyA″ most likely mediate the VLFR and HIR, respectively, whereas this may not hold for phyA over- and underexpressors (see discussion in [76]).

Besides the above phyA mutants with the complete block of the VLFR or HIR, there are modifications in the phyA molecule increasing or decreasing the strength of these responses. Of particular interest are those which relate to the natural modifications of phyA (autophosphorylation) or to its mechanism of action (kinase activity)—phyA is known as a phosphoprotein and a light-regulated kinase [141,142]. The photoreceptor is autophosphorylated at serines 8 and 18 (in oats), and this serves as a means regulation its functional activity [143,144,145]. Light-induced phyA phosphorylation modulates its interaction with transduction chain partners—a phosphorylated phyA form associates with the COP1/SPA1 complex in the cell nucleus, whereas underphosphorylated phyA predominantly associates with the intermediates FHY3 and FHY1 [146]. Mutations at Ser8Ala and Ser18Ala in oat phyA expressed in *Arabidopsis*, at the sites involved in the phyA autophosphorylation bring about hypersensitivity to FRc and FRp, which is interpreted to result from the higher light stability of the mutated phyA [145]. There were no significant changes in the phyA′/phyA″ ratio in this *Arabidopsis* line, suggesting that the effect is not connected with changes in their content but, rather, in their higher stability [147].

A decrease and increase in photoresponses to FR—both the HIR and VLFR—are also observed upon modifications in the kinase activity of (oat) phyA (expressed in transgenic *Arabidopsis*) when the kinase activity of phyA is decreased or increased, respectively [148,149]. The intrinsic kinase activity of phyA is, thus, necessary for the regulation of the components of the transduction chain—PIFs, COP1 and SPA [20,65,150,151].

The strength of phyA photoresponses, their sign and even their mode depend on plant species and their organ/tissue used. This follows, in particular, from the regulation of the key process of photomorphogenesis—protochlorophyllide (Pchlide) accumulation. The regulatory effects of phyA, the VLFR and HIR, are experimentally observed as (1) potentiation of greening—the reduction in the lag of chlorophyll synthesis by short (hours) periods of FR before the transfer of a seedling to white light [49,152] and (2) blocking of greening by prolonged FR (days) illumination of a seedling before transfer to white light [133,153,154]. A dominating view in the literature is that the regulation of the active Pchlide^655^ accumulation by phyA is negative [153,154]. We have shown, however, that the sign and magnitude of constant FRc (HIR) effect on the active Pchlide^655^ accumulation depend on the system under investigation [155]. In the cotyledons of tomato and *Arabidopsis* grown under FRc, a decline of [Pchlide^655^] was observed, in agreement with the data of Runge et al. [154] and Barnes et al. [133]. These effects on the dicotyledons were supported by experiments on monocot rice and its mutants deficient in phyA, phyB, or phyA and phyB [80]. FRc brought about a steep decline of inactive Pchlide^633^ and Pchlide^655^ in the WT plant and also its phyB mutant; pulsed FR (FRp) was of low effectiveness suggesting that these responses belong to the HIR in agreement with [156]. However, in tobacco cotyledons and pea leaves, and in stems of tobacco, pea, tomato, and *Arabidopsis*, a positive effect of FRc (HIR) on Pchlide^655^ was observed. The different signs of the FRc effect on Pchlide^655^ are not connected with the availability of the Pchlide chromophore [157]. In more detail, see a discussion on phyA regulation of Pchlide biosynthesis in [158].

The mode of the phyAs action in the regulation of Pchlide biosynthesis is well in line with the above attribution of the HIR and VLFR to phyA″ and phyA′, respectively. According to Kneissl et al. [50], the rice mutant phyA SA (expressed in *phyB* or *phyAphyB Arabidopsis*) was considerably less efficient than the WT rice phyA in the Pchlide biosynthesis suppression under FRp (VLFR), whereas the effect of FRc (HIR) was similar in both the lines. This indicates that phyA′′ is responsible for this HIR effect on Pchlide and phyA′ for the VLFR because phyA SA is represented primarily or exclusively by the phyA′′ species (i.e., lacking phyA′) [86]. Experiments on Δ6–12 phyA of oat expressed in *Arabidopsis* have shown, on the other hand, that it was hyperactive for the FRp blocking of greening upon transfer to W in *Arabidopsis* (VLFR), whereas the effect of FRc (HIR) was reduced compared with the full-length (FL) phyA [123]. Trupkin et al. [159] have similarly shown with the use of a homological system—*Arabidopsis* with the Δ6–12 deletion in *Arabidopsis PHYA*—that the 6–12 aa. region is dispensable for the VLFR but is necessary for the HIR. These data can be explained as a manifestation of the functions of the phyA′ type—the dominating or the only phyA species present in the transgenic *Arabidopsis* expressing Δ6–12 (oat) phyA [124].

Finally, an emerging important theme is the key role of plant hormones on phyAs state and their actions. In general, there is a very close connection between phyA functions and the hormonal status of the plant (see reviews [160,161,162,163,164,165,166]). We investigated the effects of the hormone jasmonic acid (JA) on the phyAs and their functions [167,168]. JA controls different aspects of plant growth and development, including inhibition of seed germination and root growth and stimulation of degradation of chloroplast proteins and leaf senescence (for review on JA see [160,169,170,171]). Experiments with rice mutants (*hebiba* and *cpm2*) lacking JA have clearly shown that JA determines the sign of the phyA regulatory effects. Reversion of the sign of the phenomenological effects was observed in phyA regulation of (1) growth responses, (2) Pchlide accumulation and (3) phyAs content and proportion. The growth of coleoptiles and seminal roots in WT rice is inhibited by the VLFR conditions, whereas for mesocotyls, by the HIR [172,173]. In our experiments, WT coleoptile growth was elevated in the dark and efficiently inhibited by R and FR, whereas mutant coleoptiles were arrested in growth if they remained in the dark but expanded rapidly upon illumination [167]. The reversion of the sign of the FRc effects was also detected on Pchlide^655^ accumulation in the mutant *hebiba*. The content of Pchlide^633^ and Pchlide^655^ in the dark was higher in the mutant pointing to the inhibitory effect of the hormone on their biosynthesis in the WT. Pulsed FR light (VLFR) stimulated it in both the wild type and the mutant, whereas constant FR (HIR) inhibited it in the WT and stimulated in the mutant. This demonstrates the dependence of the sign of the effect on the mode of FR action—VLFR or HIR—on the protochlorophyllide biosynthesis [167]. A similar anomaly of the phyA action under FR was observed in *hebiba* with regard to phyA regulation of its own state. In the dark, phyA content and the phyA′/phyA″ ratio were the same in WT and *hebiba*. Under FRc, [phyA] dropped down in WT seedlings and the phyA′/phyA″ balance shifted towards phyA″ whereas in the mutant, the [phyA] decline was less pronounced and there was no change in the phyA pools’ balance. This suggests that JA in darkness does not affect the rate of phyA synthesis and its differentiation into the phyAs subpools phyA′. Under the light, there is an interaction of the two processes: (i) autoregulation of phyA synthesis without changes in the phyAs ratio (observed both in WT and *hebiba*); and (ii) light-induced destruction of phyA primarily in the phyA′ form (only in WT).

Our recent experiments on *hebiba* and *cpm2* [168] have confirmed the sign reversal of the photoresponses. In particular, phyA suppresses root growth under FRp in the mutants but not in the WT. They also have shown that the manifestation of a photoresponse depends on the illumination conditions and on the age of the seedlings. For instance, the coleoptiles of the WT and the mutants remained unresponsive to all the light regimes. This was explained in agreement with Shimizu et al. [173] and Xie et al. [174] by the earlier age of the seedlings as compared to that in [167]. A somewhat different picture was also observed with regard to Pchlide biosynthesis. The proportion of the two Pchlide species was rather conserved, it varied within a very narrow limit. In the WT, the FRc and FRp effects on their content were insignificant, whereas in the mutants FRp was inhibiting and FRc stimulating. This is at variance with the data on the WT rice in [80], when FRc brought about a complete block of [Pchlide^655^] and a considerable decline in [Pchlide^633^], and on *hebiba* in [167], when the FR effects were different for the two Pchlide species. In the *hebiba* mutant, the reversion of the sign of the FRc effect was also observed (see above). This variability of the FR effects on Pchlide synthesis even in the same plant (rice) suggests their dependence on the physiological status of the plant, possibly, on its age (similarly to the variations in growth responses, see above). The phyA can differentially affect the biosynthesis of Pchlide under the VLFR and HIR conditions, and JA counteracts this action in WT. The suppression of the phyA action by JA may include phyA destruction (see above), and the modulation by JA of the level of the phyA transporters into the nucleus—FHY1 and FHL [175]. In general, our data on phyA regulation of Pchlide biosynthesis in the JA mutants are in line with the notion that the signals from phyA and JA are mutually antagonistic (see reviews [158,161]). Collectively, our data on JA mutant features suggest that JA reduces the phyA functional activity primarily in its phyA′′ form mediating the HIR (see the discussion on the JA and phyA interaction in [68,134]).

## 7. Action of phyA in the Cytoplasm

A number of early research indicate that there are biophysical events initiated by phytochrome in etiolated plant cells, such as modulation of ion flux and electric potential across plasma membranes. The time scale of these processes is within seconds and minutes, and it is evident that they cannot be connected with the nuclear transduction chain signalling from phytochromes via activation of photoresponsive genes, which takes hours and even days depending on the response. The existence of these membrane effects implies specific localization of phytochrome. According to Roux [99], there may be a variety of transduction chains emanating from phytochromes functioning in different cellular micro-environments. Furuya [10] points to the cytoplasm, beside the nucleus, as a site of phytochrome action.

Direct experimental evidence that phyA acts in the cytoplasm was obtained with the use of mutants with a blocked process of nuclear-cytoplasmic phyA partitioning. Certain photophysiological responses (such as R-enhanced phototropism, abrogation of gravitropism, and inhibition of hypocotyl elongation in blue light) are seen in the mutants lacking FHL and FHY1, which are necessary for the appearance of the nuclear-localized fraction of phyA after its light activation [25,26,27,28,29]. The *fhl/fhy1* mutant retained the phyA mediated enhancement of blue light-induced phototropism [26]. This enhancement is observed upon red pre-irradiation of dark-grown *Arabidopsis* seedlings [176]. This effect involves the modulation of phototropin, which is predominantly localized at the plasma membrane [177,178]. The integration between the phytochrome A and phototropin regulatory pathways involves Phytochrome Kinase Substrate 1 (PKS1)— a protein associated with the plasma membrane, which interact with both PHYA and PHOT1 [179,180]. The effects of phytochrome on hypocotyl growth [176] involve an integral membrane-bound protein associated with auxin transport [181] (see review [182]). The control of translation of PORA mRNA is also mediated by cytoplasmic phyA: phyA in the active Pfr state binds to PENTA 1 (PNT1) and represses the translation of mRNAs [28,29]. The phyA effects are likely to be realized through direct interaction of the photoreceptor with plasma membrane protein partners. Ion fluxes across the plasma membrane may participate in light-invoked signal transduction [183].

Recent findings widen the range of cytosolic regulatory events. It was shown that modification of transcriptional processes may also take place in the cytoplasm. The Pfr form present in the cytosol interacts with the cytosolic protein PENTA1 (PNT1) and inhibits the translation of protochlorophyllide reductase (PORA) mRNA. The light-dependent recruitment of phyB and phyA leads to the translational inhibition of PORA mRNA. These results demonstrate that phytochromes transmit light signals to regulate not only transcription in the nucleus through PIFs, but also translation in the cytosol through PNT1 [28]. Schwenk and Hiltbrunner [184] have found that phyA participates in the control of translationally halted mRNAs that are stored in processing bodies—cytosolic RNA granules containing mRNAs [185]. Upon FR, these bodies are disassembled, and stored mRNAs are released and contribute to plant’s adaptation to the light environment. The fact that phytochrome A is sufficient and necessary for the FR light-induced disassembly of processing bodies are shown with the use of the *fhl/fhy1* mutant defective in the light-induced nuclear import of phyA. On the contrary, the authors could not observe the effect in the *Arabidopsis* transgenic line containing exclusively the nuclear–localized phyA [127].

Phytochrome participates in regulating phototropic responses and primary root elongation growth [186,187,188]. Shin and co-workers [189] have followed cytoplasmic phytochrome action in root development. They detected a GTPase activator protein PIRF1 (phytochrome-interacting ROP guanine–nucleotide exchange factor (RopGEF 1) that localized and interacted with phytochromes in the Pr form in the cytoplasm in the dark. It remained there even after Pr was photoconverted to Pfr. It functions as a light-signaling switch regulating root development through the activation of ROPs (Rho-like GTPase of the plant) in the cytoplasm. The Pr form of phytochrome A enhanced the RopGEF activity of PIRF1, whereas the Pfr form inhibited it. PIRF1 was localized in the cytoplasm and bound to the phytochromes in darkness but not in light. The authors came to the conclusion that PIRF1 is a negative regulator of phytochrome-mediated primary root development and that phytochrome and ROP signaling are interconnected through PIRF1 in regulating root growth and development in *Arabidopsis*. The authors also underline that unlike most other phytochrome-interacting proteins, PIRF1 interacted specifically with an N-terminal domain of both phyA and phyB. Participation of phyA is also suggested by the fact that the production of GTP by an NDPK2 enzyme was specifically activated by the Pfr form of phyA [190] implicating phyA in the above effects.

Given that the association of phyA with cytosolic proteins, in particular, with PKS1, is required for the above cytosolic photoregulation effects, we may assume that the most likely candidate for this association is amphiphilic phyA″. Its affinity for protein association makes phyA″ (its membrane-associated fraction phyA′′m) the most likely candidate for these functions. The other fraction of phyA″, which is not bound to the membrane (protein), is involved together with the whole pool of the soluble phyA′ in the nuclear regulation events. In this connection, it is interesting to note that PKS1 and PKS2 are intimately related to the state of the phyA pools in the cell [87]. We found that the *Arabidopsis pks1pks2* double mutant has a much greater proportion of phyA″ at the expense of phyA′ (Figure 9). It is tempting to hypothesize that the phyA′′m pool could be also responsible for the fast photoregulation effects in the cytoplasm, such as modulation of ion transport, electric potentials, and cytoplasm fluidity (Figure 12). Thus, the difference in the hydrophilicity/hydrophobicity of the two phyA pools (see above) may account for the specificity of phyA functioning in the cytoplasm.

## 8. Regulation of the phyAs Content and Their Balance in the Dark and in the Light

Light has specific impacts on different plant tissues and organs during the process of photomorphogenesis and throughout various stages of the plant life cycle—promoting growth in some of them and inhibiting expansion in others. This is connected with the character of spatial localization of photoreceptors and with distinct signalling cascades downstream of the activated photoreceptors in distinct tissues. In particular, the spatio-temporal phytochrome responses are considered to be central to coordinated plant growth, development and metabolism [63]. The detection of the two phyA populations makes the functional behaviour of the photoreceptor even more complicated. We have firmly documented that the content of the two phyAs and their ratio are strongly dependent on the plant species, their organs and tissues used and stage of the plant’s development. In young etiolated seedlings, there is a domination of phyA″; however, along with the accumulation of total phyA, phyA′ becomes the major species whereas the concentration of phyA″ reaches saturation (Figure 4). In general, phyA′ is a labile species dominating in growing etiolated tissues, whereas phyA″ is more stable with a high proportion in resting tissues [67,71,137].

The mechanisms determining this complex behaviour of phyAs remain unclear, although there are indications that the phosphatase/kinase equilibrium in the cell and cytoplasmic pH affect it [71,88]. The agents suppressing phosphatase activity and shifting the phosphatase/kinase equilibrium towards kinases (okadaic and cantaridic acids—inhibitors of phosphatases of the PP1 and PP2A types, and NaF—phosphatase inhibitor of a broad spectrum) brought about the increase in the phyA″ content relative to that of phyA′ due to the elevated destruction of the latter or its conversion into phyA″. Interestingly, the agents used revealed their organ specificity—the acids were active in (maize) coleoptiles and inactive in roots, whereas NaF, *vice versa*. This supported the notion of the specificity of the phyA state in the two organs [191]. The phyA′/phyA″ ratio was found to be connected with pH: it qualitatively correlated with the pH in maize root tips during their growth—fast increase in phyA′ at the moment of radicle protrusion up to 2 mm length (the first stage), then dominance of phyA′′ (the roots length of ca 5 mm, stage 2) and finally, at stage 3 (the tips of the roots of 10–25 mm length)—again domination of phyA′ over phyA′′. In extracts from maize coleoptiles, a bell-shaped dependence of the ratio with the maximum at around 7.5–7.6 characteristic for the cell cytoplasm was observed. These data are in good qualitative agreement with the effects of pH on roots. Root tips soaked in buffer solutions of different pH revealed considerable variations in the phyAs ratio—steep drop of the phyA′ proportion at pH 4.9 and 8.0, whereas at pH 6.4 it was maximal characteristic for intact roots. Similar effects were observed on coleoptiles. It should be noted that these pH effects could not be the consequence of direct protonation/deprotonation and/or phosphorylation/dephosphorylation of the phyA molecule as revealed in the experiments on heterologously expressed and purified *Arabidopsis* phyA, a full-length (phyA FL) and N-terminal sensor domain fragment (phyAΔ3), expressed in *P. pastoris* and *E. coli*. This phyA belonged to the phyA′′ type and was not affected by phosphatases and pH variations [71]. Thus, we may speculate that the above effects on the phyAs ratio in the cell are intimately connected with the regulation of the complex processes of phyAs biosynthesis, interconversion and destruction.

These effects taking place in the dark are superimposed by even more pronounced effects seen after pre-illumination. They are clearly distinguished into those induced by red and far-red light, which may reflect the specificity of the phyA action as a mediator of the far-red light. After R pre-illumination, fast destruction (tens of minutes) of the major phyA′ pool is observed, whereas phyA″ remains relatively unchanged [79]. This results in a sharp shift in the phyA′/phyA″ ratio towards phyA″ suggesting that under light conditions it is a dominating species. It should be mentioned that short (min) red saturating pre-illumination followed by saturating far-red illumination of the sample reversing the cycle of Pr-Pfr-Pr transformations does not affect the phyAs ratio and their spectral properties [74]. This indicates that phyA in the Pr* form, i.e., Pr cycled through Pfr, is identical in this regard to the initial dark-adapted phyA. A quite different picture is seen in plants grown under FR: instead of preferential phyA′ distraction there is a total phyA decline without violation of the phyAs equilibrium characteristic of phyAs in the etiolated state (in pea) or with a relatively small shift towards phyA″ (in maize). Judging by the fact that the de-etiolated *lip* mutant (of pea) without pre-illumination reveals similar properties as the FR pre-illuminated WT pea, i.e., total phyA decline without phyAs ratio shift [192], we may conclude that this FR effect is primarily a consequence of the negative feedback of the phyA autoregulation of its own synthesis during de-etiolation. In this process, its destruction in the phyA′ form is suppressed. Of interest is the delayed effect of germination-inducing R pre-illumination of *Arabidopsis* seeds on the state of the two phyAs in growing seedlings [86]. It was stimulating for the formation of phyA′ as compared with the seedlings without such a light pretreatment. This suggests that, besides the phyA′ destruction, R light may stimulate phyA′ formation, possibly modulating seedling’s development, and that the phyA differentiation into the two subspecies is a light-regulated process. Collectively, these data suggest that we have to consider the variations in the content and proportion of the phyA pools as a part of the very complex process of phyA fine-tuning.

## 9. Conclusions

The aim of this review was to present the concept that phyA in plants is represented by two distinct species possessing different physicochemical properties, and that their existence in the cell can explain the diverse types of photoresponses it mediates. Detection of phyA fluorescence in-situ and the development of the sensitive fluorescence method made it possible to assay phyA in detail in its native state in the cell. The pigment was characterized by its content in tissues, fluorescence emission and excitation (absorption) spectra, and activation and kinetic parameters of the photoreaction. With the use of this method, experiments along several major lines were carried out and each of them pointed to the heterogeneity of phytochrome in situ. First, it was clearly shown that all the phytochrome parameters greatly vary depending on plant species/organ/tissue, their developmental state, and the effect of environmental factors. Experiments with phytochrome mutants provided compelling evidence that these variations cannot be explained by the presence of phyB and other minor phytochromes and pointed to the intrinsic heterogeneity of phyA. This heterogeneity was shown to reflect the existence of (a) chemically distinct species and (b) their conformers. Two phyA states appear in the cell as a result of post-translational modification—the first is the major, light-labile, photochemically active at Tc and hydrophilic (phyA′), and the second, the minor, saturable by its content, relatively light-stable, inactive at Tc and amphiphilic (phyA″). Both of them are full-length products of one and the same *PHYA* gene and reveal normal photochemical activity at Ta. The phyA″ is present in the cell in a water-solved state and in association with the membrane (protein) (designated phyA″m). Within phyA′, at least three conformers differing by activation parameters of the photoreaction were detected. Structural, steady-state and time-resolved investigations of cyanobacterial phytochrome Cph1, an adequate model of plant phytochromes, revealed its heterogeneity too— the existence of conformers of the pigment—and explained it in terms of concrete chromophore-apoprotein interactions. The three states of phyA, phyA′, phyA″ and phyA″m, are believed to be responsible for the complex phyA action: phyA′ and phyA″ were shown to differ by the mode of the nuclear-cytoplasmic partitioning and mediate the VLFR and the HIR, respectively. The phyA″m is hypothesized to remain in the cytoplasm upon illumination and participate in the cytoplasmic photoregulation effects. The character of the phyAs photoresponses—their mode, strength, and sign—depend on the plant (wild type, phytochrome mutants, over- or underexpressors) and their organ/tissues under investigation. A closely related important theme is the interaction of phyA with plant hormones. From experiments with mutants lacking the hormone jasmonic acid (JA) participating in plant’s defence, it was shown that JA determines the sign of the phyA responses (regulation of seedling growth, Pchlide biosynthesis, phyAs content, and balance), and, in general, JA antagonizes the action of phyA. One of the main questions—the exact mechanism of the phyA post-translational modification converting phyA″ into phyA′—remains, however, open for further experimental solution.

## Figures and Tables

**Figure 1 ijms-24-08139-f001:**
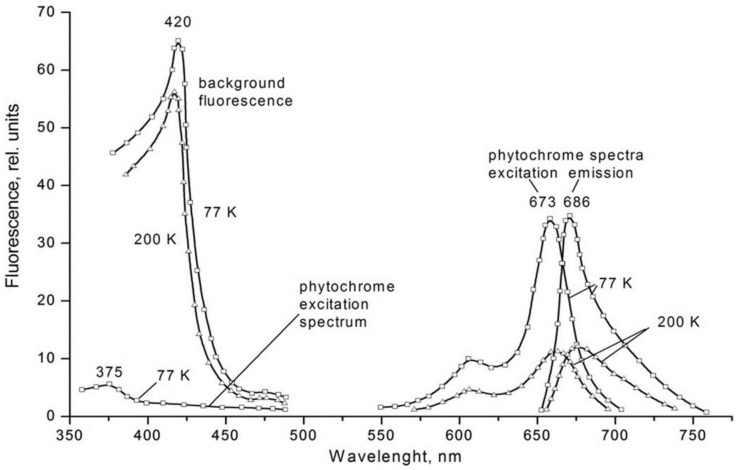
Fluorescence emission (excitation wavelength, λ_ex_ = 630 nm) and excitation (emission wavelength, λ_em_ = 700 nm) spectra of phytochrome in the Pr form in stems of etiolated pea seedlings at 77 K and 200 K. Excitation spectrum of phytochrome with the maximum at 375 nm was calculated in the region 350–500 nm with due consideration of green background fluorescence with an excitation maximum at 420 nm. Procedure of the calculation is based on different temperature dependences of phytochrome and background fluorescence. From [42,43].

**Figure 2 ijms-24-08139-f002:**
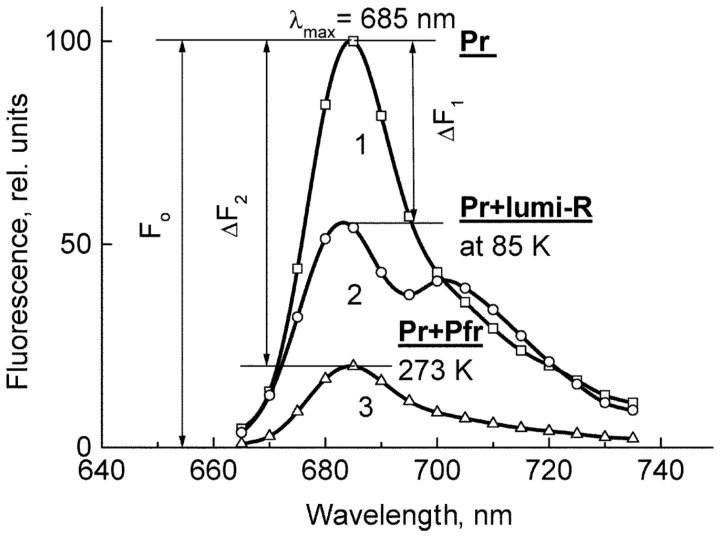
The low-temperature fluorescence method for phytochrome in-situ assay. To characterize the pigment, three fluorescence spectra of phytochrome were measured at 85 K: (1) in etiolated tissues (of wheat in this figure) when all phytochrome is in its Pr form; (2) in the same sample after saturating red illumination at 85 K partially converting Pr into lumi-R, the first stable photoproduct lumi-R; and (3) in the same sample after thawing at 273 K, saturating red illumination converting Pr into Pfr and freezing again at 85 K. Four major parameters were obtained from these spectra: position of the emission spectrum (λ_max_); total phytochrome content proportional to the fluorescence intensity ([P_tot_] ≈ F_0_); extent of the Pr lumi-R photoconversion, equal to the relative fluorescence decline (γ_1_ = ∆F_1_/F_0_) and characterizing the initial photoreaction; and the extent of the Pr Pfr photoconversion (γ_2_ = ∆F_2_/F_0_), characterizing the whole phytochrome cycle. From [67,68].

**Figure 3 ijms-24-08139-f003:**
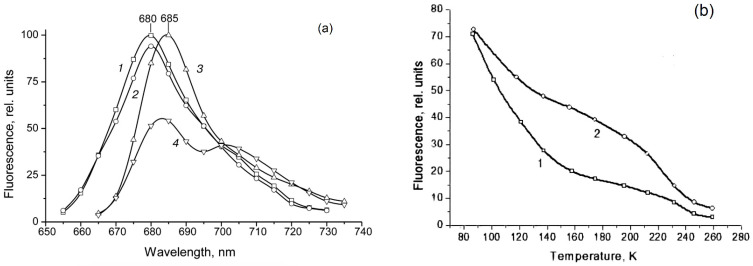
Dependence of phytochrome fluorescence and photochemical characteristics on plant organ/tissues. (**a**) Fluorescence emission spectra (85 K, λ_ex_ = 632.8 nm) of phytochrome in root tips (λ_max_ = 680 nm) (1, 2) and coleoptiles tips (λ_max_ = 685 nm) (3, 4) of etiolated wheat seedlings measured immediately after darkness when all the pigment is in the Pr state (1, 3) and after saturating red illumination (actinic wavelength, λ_a_ = 632.8 nm) partially converting Pr into lumi-R, the first product stable at low temperatures (the state of photoequilibrium between Pr and lumi-R) (2, 4). Note the difference in the position of the spectra (λ_max_) and the extent of the photoconversion Pr→lumi-R (γ_1_) measured as a relative decline of the intensity in the maximum, which are 680 nm and 0.05, respectively, for roots and 685 nm and 0.46 for coleoptiles. The spectra were not corrected for the spectral sensitivity of the spectrofluorometer. From [69]. (**b**) Temperature dependence of fluorescence intensity (λ_ex_ = 650 mm, λ_em_ = 686 nm) of phytochrome in the cells stems (1) and roots (2) of 5-day-old etiolated pea seedlings. From [70].

**Figure 4 ijms-24-08139-f004:**
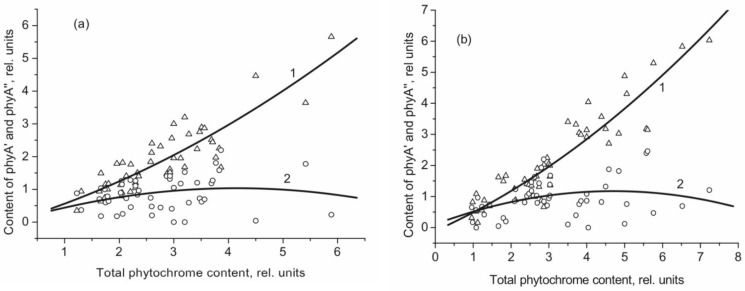
Dependence of the content of the two native phyA types, phyA′ and phyA′′ (open triangles and open circles and their polynomial fits 1 and 2, respectively) on the total concentration of phyA, ([P_tot_]), in etiolated maize roots (**a**) and coleoptiles (**b**) at different stages of their development (summary data of different samples of the Kubanskaya var). From [71].

**Figure 5 ijms-24-08139-f005:**
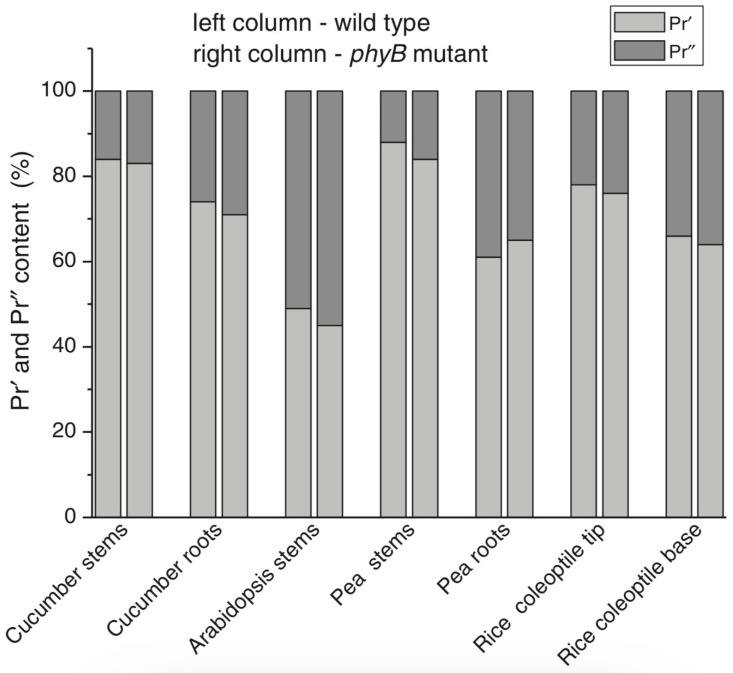
Relative content of the two phytochrome types, Pr′ and Pr″, differing by spectroscopic, photochemical and phenomenological properties (see Table 1), in etiolated dicotyledonous and monocotyledonous plants and their phyB-deficient mutants. Close similarity in the content of Pr′ and Pr″ between the wild type plants and the mutants strongly suggest that there exist two phyA species, phyA′ and phyA″, with the properties of Pr′ and Pr″ respectively (modified from [77,78,79,80]).

**Figure 6 ijms-24-08139-f006:**
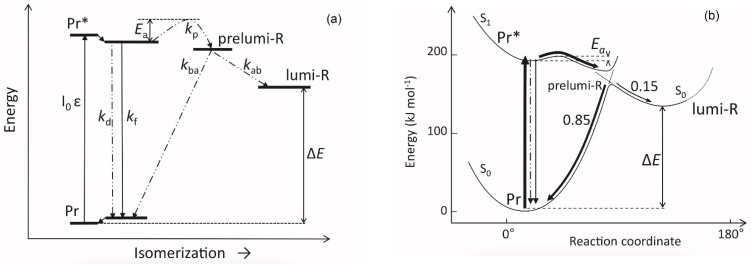
Energy level schemes of the photoreaction of the initial red-absorbing form (Pr) (Pr* is the excited state of Pr) into the first photoproduct (lumi-R) stable at low temperatures via a short-lived unstable orthogonal intermediate (prelumi-R) suggested for plant phytochrome (**a**) and for the cyanobacterial phytochrome Cph1 (**b**). At 85 K and saturating red illumination (λ_a_ = 632.8), there is a photoequilibrium between Pr and lumi-R determined by the rates of the forward (Pr→lumi-R) and reversed (lumi-R→Pfr) photoreactions. The activation barrier in the excited state, *E*_a_ for Pr and *E*_a_′ for lumi-R, determines the photochemical properties of Pr and lumi-R and the extent of the Pr photoconversion to reach a photoequilibrium with lumi-R at low T (γ_1_ varies for different Pr species from 0 to 0.5). At ambient temperatures, this barrier is easily overcome and the extent of the Pr→Pfr photoconversion γ_2_ remains relatively constant, 0.75–0.85. (From [14,70]). (**b**) Hypothetical potential energy curves and quantum yields of the Pr photoreaction in Cph1 based on the scheme in (**a**). From [81].

**Figure 7 ijms-24-08139-f007:**
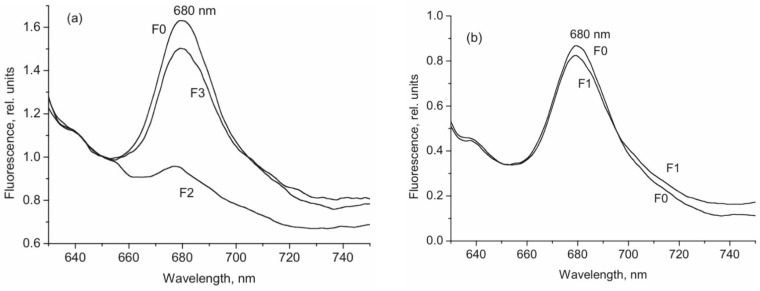
Fluorescence emission spectra (λ_ex_ = 630–633 nm) of heterologously expressed phyA in *P. pastoris* at 77 K. (**a**) The spectra of phyA with endogenous chromophore in *P. pastoris* labeled as F_0_, F_2_ and F_3_ were taken after actinic FR-R-FR illumination at room temperature, respectively. Two cycles of such phototransformations were performed on one and the same sample proving that the spectrum belongs to phyA. (**b**) Determination of γ_1_ for phyA in *P. pastori*s: the spectra F_0_ and F_1_ (after actinic R illumination at 85 K) were taken at 77 K. The fact that spectrum F_0_ is close to spectrum F_1_, that is, the γ_1_ value approaches 0, suggests that phyA in *P. pastoris* belongs to the phyA′′ type. From [71].

**Figure 8 ijms-24-08139-f008:**
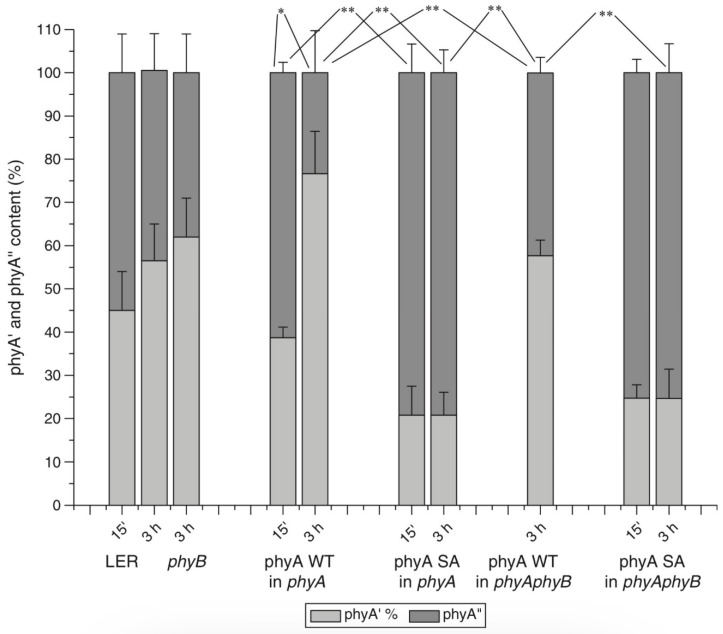
Proportion of the two phytochrome pools, phyA′ and phyA″, in etiolated *Arabidopsis* plants of the different lines. Significantly different pairs of data (revealed by Fisher’s *t*-test for 6–12 independent measurements) are indicated: *, *p* < 0.05 for the same plant lines after brief (15 min) and prolonged (3 h) white light germination-inducing pre-treatment and **, *p* < 0.05 between different plant lines taken at the same pre-illumination time, 15 min or 3 h. From [86].

**Figure 9 ijms-24-08139-f009:**
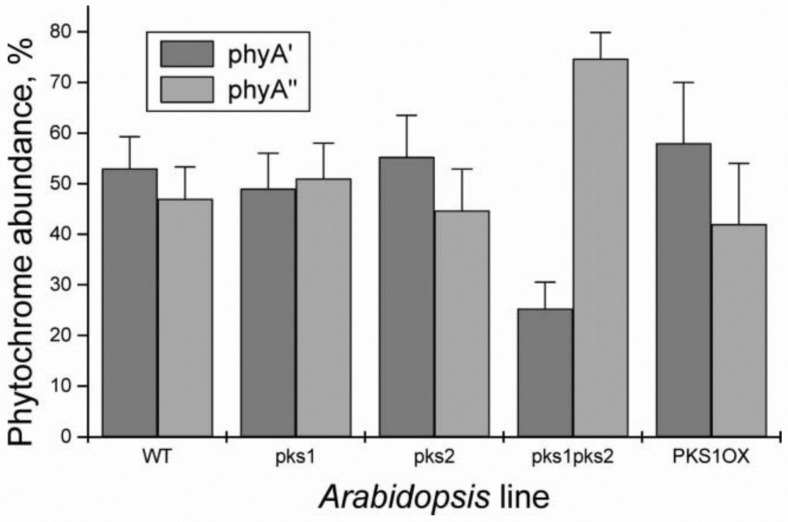
Relative content of phyA′ and phyA″ (%) in etiolated seedlings (without cotyledons) of different *Arabidopsis* lines (left column: phyA′; right column: phyA″). From [87].

**Figure 10 ijms-24-08139-f010:**
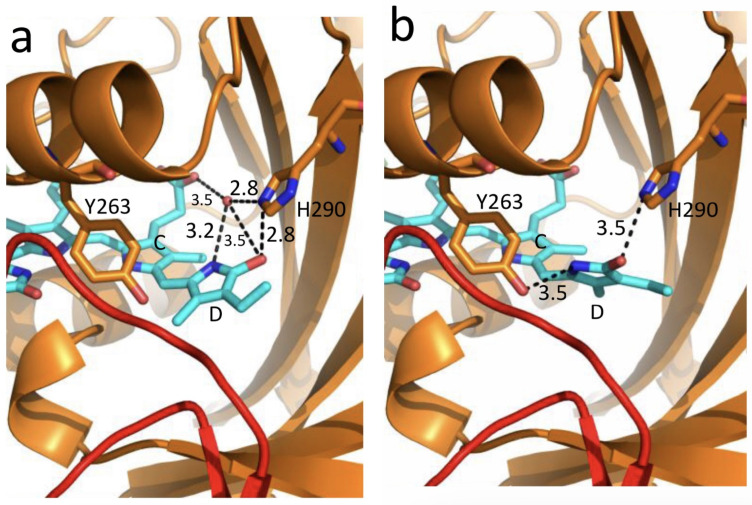
The chromophore D ring environment of Cph1 and its changes in the photoreaction. (**a**) 2VEA structure of Cph1 Pr [90] showing the interactions of the pyrrole N24 and C19 carboxyl oxygen; (**b**) hypothetical structure of prelumi-R at C15 = C16 isomerization angle of 117°. From [81] modified).

**Figure 11 ijms-24-08139-f011:**
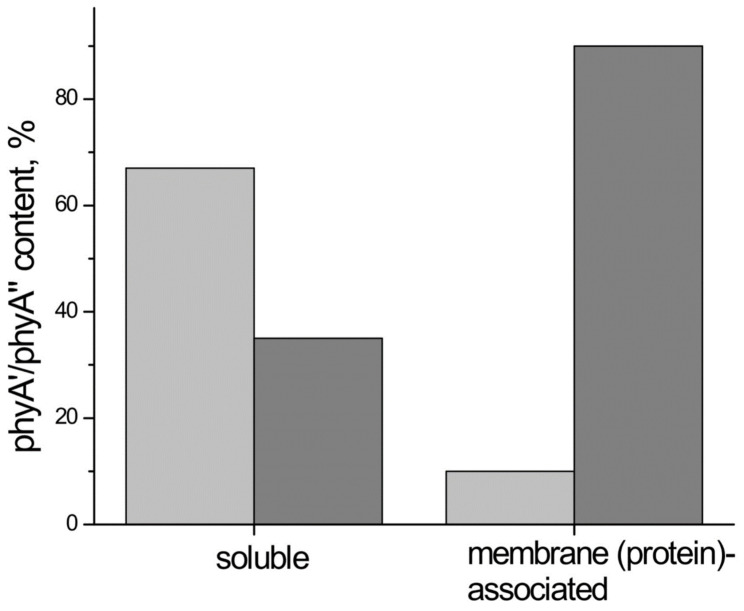
Proportion of phyA′ (light gray) and phyA″ (gray) in water-soluble and membrane-associated fractions of the intact 124 kDa maize phytochrome A in vitro. The error (SD) of the phyA content determination is ca ± 5%. From [109].

**Figure 12 ijms-24-08139-f012:**
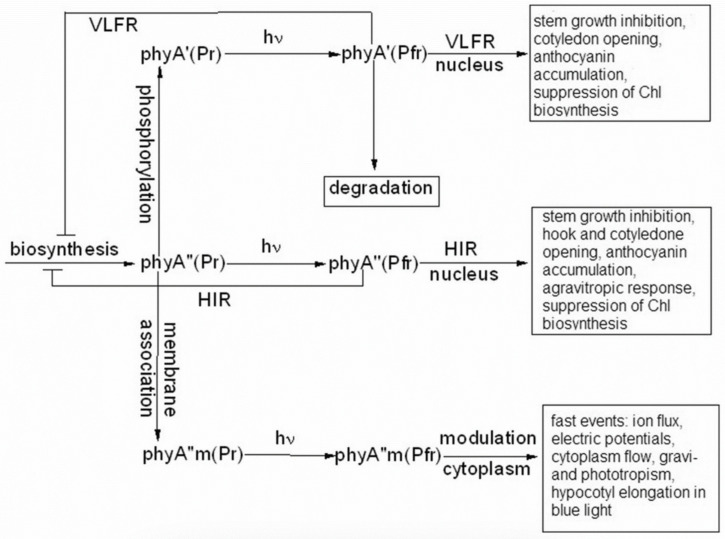
Working scheme of the state and functions of the native phytochrome A pools in etiolated seedlings. De novo synthesized phyA in germinating seeds and growing seedlings is initially in the phyA′′ form, which possesses amphiphilic properties and is present in the cell in water-soluble and membrane-(protein-) associated (phyA′′m) fractions. In darkness, phyA′′ is converted into the water-soluble phyA′ form, possibly, via serine phosphorylation at the N-terminus of the molecule. Upon illumination, the water-soluble phyA′ and phyA′′ are transported into the nucleus forming two different types of nuclear speckles and inducing different modes of photoresponses, the VLFRs and HIRs, respectively. phyA′′m in the Pfr form remains in the cytoplasm and initiates regulation processes there. Pointed arrows indicate the stimulation effects; blunt-ended arrow, the inhibitory effects. From [134].

**Table 1 ijms-24-08139-t001:** Two phenomenological phytochrome A types in mono- and dicotyledonous plants—Pr′ (phyA′) and Pr″ (phyA″).

Parameter	Phytochrome Type	
	Pr′ (phyA′)	Pr″ (phyA″)
Position of emission/absorption maxima, λ_max_, nm	Longer wavelength (685/672)	Shorter wavelength (680/667)
Half-band width, nm	22–24	30
Extent of Pr→lumi-R conversion at 85 K, γ_1_	0.49 ± 0.03	≤0.05
Activation barrier in excited state, *E*_a_, kJ/mol	≤1	≥10
Extent of Pr→Pfr conversion at 273 K, γ_2_	0.80–0.85	0.75
Light lability	Light labile	Relatively light stable
Hydrophilicity/Hydrophobicity	Water–soluble	Ambiquitous—soluble and membrane-(protein-) associated
Content in etiolated tissues	Major, variable	Minor, saturated, conserved

## Data Availability

Not applicable.

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
