# Peer review of "Two Distinct Molecular Types of Phytochrome A in Plants: Evidence of Existence and Implications for Functioning"

_ijms, 2023, doi:10.3390/ijms24098139_

Round 1
Reviewer 1 Report
This manuscript submitted by Prof. V. Sineshchekov describes about two distinct types of phytochrome A (phyA) in plants. The author worked for a long time with this issue, so reviewed the diverse aspects of phyA action with light-labile/soluble (phyA¢) and relatively light-stable/amphiphilic (phyA¢¢) types thoroughly in this review. Overall, the manuscript was well-written and informative to be published.
Here are some comments to improve this manuscript.
It is necessary to arrange Figures/Table for better reading by adjusting their sizes and positions. In addition, please define abbreviations at the first time (For example, Ta, Tc, etc.). Moreover, it needs to be careful in using Figures that have been previously published. For example, the positions of (a) and (b) in Figure 3 are not consistent [i.e., kind of copy/paste from PDF file (Figures 5, 6, 8 and 12, too); better to use original figures]. Since this is a review paper, experimental data may be better to be presented as drawings or simpler formats, if it’s possible.
There are many references missing in the Reference list. For examples, in pages 2, 3, 11, 17, 18, 24, and 25, the following references could not be found: “Hiltbrunner et al., 2006”, “Rösler et al. 2007”, “Jaedicke et al. 2012”, “Hughes, 2013”. “Lin et al., 2022”, “Han et al., 2022”, “Sokolova et al., 2012”, “Hahm et al., 2020”, “Yanovsky et al., 2002”. “Han et al., 2010”, “Liu and Wang 2020”, “Whippo and Hangarter 2003”, “Schwenk and Hiltbrunner (2022)” and more. Therefore, it is necessary to match the references in the text with the list.
This review is a quite long paper (32 pages). Probably, some parts can be described more clear, concise, and compact. For example, in pages 24-25, the description of PNT in two paragraphs could be combined, and so on. For readers, please try to reduce the length of manuscript by focusing on the aims in the Conclusions.
Minor comments:
Page 2, change comma to semicolon in “Hiltbrunner et al., 2006; Rösler et al. 2007;”
Page 3, check the unit in “(<10-7 M/l)” [would be “<10-7 M”]
Page 4, in the legend of Figure 1, need to check “Fluorescence emission (λex = 630 nm) and excitation (λem = 700 nm)”. [should be emission = λem & excitation = λex]
Page 8, revise “Ea Ea”.
Page 9, check the unit in “Ea = 12.5–17.5 kJ mol#-1”. [#-1 should be 10 to the -1]
Page 13, revise “Cph1 (3.0–6.5 and 12.0–17.5 kJ mol-1 for the two Cph1 species (Sineshchekov et al., 2014b)”. [insert “(” & “)” correctly]
Page 22, revise “chro#mophore”. [remove “#”]
Page 25, “pant” to “plant”.
Pages 25-26, need to check fonts.
Page 27, “Conclusion” to “Conclusions”.
Page 28, the format is not correct. Check the references.
Throughout the manuscript, please revise more carefully by removing unnecessary spaces and “-”.
Author Response
Please see the atachment.

Reviewer 2 Report
Review of the article by V. Sineshchekov: "Two Distinct Molecular Types of Phytochrome A in Plants: Evidence of Existence and Implications for Functioning "
The author's review article deals with the consideration of the question of the presence in plants of various types of phytochrome A having different physico-chemical properties and that their presence in the cell can explain different types of photo responses.
It has been shown that phyA mediates three types of reactions — very low and low frequency responses (VLFR and LFR) and responses with high radiation intensity (HIR). phyA is the only light receptor in the far-red region of the spectrum responsible for the survival of plants in dense vegetation conditions, where light is enriched with a far-red component.
The detection of phyA fluorescence in situ and the development of a sensitive fluorescence method made it possible to analyze phyA in its native state in a cell in detail. The pigment was characterized by its content in tissues, fluorescence and excitation emission spectra, as well as activation and kinetic parameters of photoreaction. Using this method, experiments were carried out in several main directions, each of them indicated the heterogeneity of phytochrome in situ. It has been shown that all the parameters of phytochrome vary greatly depending on the type of plant/organ/tissue. Two phyA states appear in the cell as a result of posttranslational modification - the first is the main, light-stable, photochemically active at Tc and hydrophilic (phyA'), and the second, minor, saturated in its content, relatively light-resistant, inactive at Tc and amphiphilic (phyA"). Both of them are full-sized products of the same PHYA gene and exhibit normal photochemical activity in Ta.
It is believed that three phyA states — phyA', phyA" and phyA"m — are responsible for the complex action of phyA: it has been shown that phyA' and phyA" differ in the method of nuclear-cytoplasmic separation and mediate VLFR and HIR, respectively.
After conducting a detailed analysis of the available literature and his own data, the author notes that one of the main issues — the exact mechanism of posttranslational modification of phyA, which turns phyA "into phyA', remains open for further experimental solutions.
I believe that the author has fulfilled the task set for himself – to consider the current state of the issue in full and this work should be accepted for publication in the journal.
Author Response
Please see the attacjment.
